# XAudit : A Learning-Theoretic Look at Auditing with Explanations

**Chhavi Yadav**                                 *cyadav@ucsd.edu UC San Diego*

**Michal Moshkovitz**                  *michal.moshkovitz@il.bosch.com Bosch Center for AI*

**Kamalika Chaudhuri**            *kamalika@cs.ucsd.edu UC San Diego, Meta AI*

**Reviewed on OpenReview:** *https://openreview.net/forum?id=gPtjyzXskg*

## Abstract

Responsible use of machine learning requires models to be audited for undesirable properties. While a body of work has proposed using explanations for auditing, how to do so and why has remained relatively ill-understood. This work formalizes the role of explanations in auditing using inspirations from active learning and investigates if and how model explanations can help audits. As an instantiation of our framework, we look at 'feature sensitivity' and propose explanation-based algorithms for auditing linear classifiers and decision trees for this property. Our results illustrate that Counterfactual explanations are extremely helpful for auditing feature sensitivity, even in the worst-case. While Anchor explanations and decision paths may not be as beneficial in the worst-case, in the average-case they do aid significantly as demonstrated by our experiments.

## 1 Introduction

The recent success of machine learning (ML) has opened up exciting possibilities for potential societal applications Bojarski et al. (2016); Aini et al. (2020); Castiglioni et al. (2021). However, this kind of widespread usage requires us to be able to audit these models extensively and verify that they possess certain desirable properties related to safety, robustness and fairness.

So far, due to complexities in the ML pipeline and the black box nature of most models, auditing has been primarily performed in a somewhat ad-hoc manner, usually by creating a separate audit testing set Soares et al. (2023). This, however, leaves open the question of what an audit really signifies. Additionally, a recent body of work has also posited that local explanations might be helpful for auditing Bhatt et al. (2020); Oala et al. (2020); Adebayo and Gorelick (2017); Watson and Floridi (2021); Zhang et al. (2022); Poland (2022); Hamelers (2021); Carvalho et al. (2019); however, in the absence of a formal framework, how or why precisely this is the case is not understood.

Recently, Yan and Zhang (2022) has made one of the first attempts at providing a formal framework for auditing and proposed principled auditors for auditing demographic parity with rigorous guarantees. However a limitation of their work is that they do not incorporate explanations into their framework. We take this line of work forward by formalizing the role of explanations in auditing.

In the context of auditing, the model-to-be-audited is held by a data scientist and not revealed to the auditor due to confidentiality reasons – this hidden aspect is what makes the task of auditing challenging and consequently, the auditor's objective is to acquire insights into the concealed model. To achieve this, the auditor queries data points to the data scientist and in turn the data scientist responds with labels and (local) explanations. This paper aims to measure whether the supplementary information derived from explanations, beyond mere labels, can effectively diminish the query complexity of auditing.

In this paper, we consider auditing 'feature sensitivity', for two hypothesis classes with different local explanation methods – linear classifiers with counterfactual and anchor explanations and decision trees with decision path explanations. We provide auditing algorithms and theoretical guarantees for these cases. Our results illustrate that explanation methods differ greatly in their audit efficacy – while worst-case anchors do not provide any additional information than predictions themselves and with decision paths total queries scale linearly with number of nodes in the tree in the worst-case, surprisingly counterfactuals greatly bring down the query complexity of auditing to a single query when considering feature sensitivity.

We empirically evaluate our proposed auditors on standard datasets. Our experiments show that unlike the worst case, 'typical' anchors significantly reduce the number of queries needed to audit linear classifiers for feature sensitivity. Additionally, our proposed Anchor Augmentation technique helps reduce the query complexity over a no-anchor approach. Similarly, our experiments on decision trees demonstrate that the average number of queries to audit feature sensitivity is considerably lower than number of nodes in the tree.

Could there be a more general auditing strategy in our framework? We next show that this is possible. By drawing on a connection between auditing and membership query active learning, we provide strategies for two critical aspects of any auditor – picking the next query and stopping criteria. We show that our general strategy leads to successful auditing when the property to be audited is *testable* – in the sense that it can be inferred based on making queries to the model.

To summarize, our contributions are as follows.

1. We formalize the role of explanations in auditing and propose explanation-based algorithms to audit feature sensitivity in linear classifiers and decision trees.
2. We empirically illustrate that unlike the worst case, in the average case, explanations help reduce the query complexity of auditing feature sensitivity significantly.
3. We propose a general strategy for auditing (and with explanations) by drawing a connection between auditing and active learning.
4. Finally, we discuss ways of dealing with an untruthful data scientist and the privacy concerns arising from auditing.

## 2 Preliminaries

The four key components involved in an ML audit are : 1) Data Scientist (DS), 2) Auditor, 3) Model to be audited and 4) Auditing Property (AP). DS is the one who holds a model which it cannot reveal for confidentiality reasons, while an Auditor is an external entity who wants to verify certain properties of the hidden model. We assume that the auditor has query access to this model through the DS. The hidden model, $\bar{h} : \mathcal{X} \times \{-1, 1\}$, belongs to a hypothesis class $\mathcal{H}$. We assume that the hypothesis class is known to the auditor. Let the instance space $\mathcal{X} = \mathcal{R}^d$.

The auditor will audit the model held by the DS for a specific auditing property that we measure quantitatively by a score function $s : \mathcal{H} \to [0, 1]$. This function has a value of zero when the property is absent in the model and a high value when the model exhibits the property to a large extent. Auditor knows the exact form of the score function but does not know its value for $\bar{h}$ (since $\bar{h}$ is hidden).

We expect an auditing property to satisfy the notion of 'testability', defined below. Testability ensures that the property of a hypothesis can be determined by making queries to instances $x \in \mathcal{X}$. Absence of testability makes it possible to have two hypotheses that agree on *every* input in the instance space, yet may have very different values of the property (for instance, robustness to distribution shifts or adversarial attacks). Auditing this kind of a property would require more information than simply queries.

**Definition 1** (Testable Auditing Property). *An auditing property defined by score function $s : \mathcal{H} \to [0, 1]$ is testable using an instance space $\mathcal{X}$ if the following holds: for any $h_1, h_2 \in \mathcal{H}$ if $h_1(x) = h_2(x) \ \forall x \in \mathcal{X}$, then $s(h_1) = s(h_2)$.*

When a query is asked to the DS, it returns the corresponding label and a local explanation. Both the DS and auditor agree upon a specific explanation method before auditing begins. Let $e_h : \mathcal{X} \to \mathcal{E}$ denote this explanation method where $\mathcal{E}$ represents the codomain of $e_h$ and $h \in \mathcal{H}$. We assume that the DS is truthful,

meaning the labels and explanations returned for $x$ are $y = \bar{h}(x)$ and $e_{\bar{h}}(x)$ respectively. In §6 we discuss what happens when this assumption does not hold.

At the end of the auditing process, the auditor responds with an answer $Y_a$ which takes values $\{\texttt{Yes}, \texttt{No}\}$. This is a random variable since both the DS and auditor can be randomized algorithms.

Next we conceptualize the auditing process as an interaction between the DS and auditor. Our protocol is as follows.

At each time step $t = 1, 2, \ldots$

1. Auditor picks a new query $x_t \in \mathcal{X}$ and supplies it to the DS.
2. DS returns a label $y_t \in \mathcal{Y}$ and an explanation $e_t \in \mathcal{E}$ to the auditor.
3. Auditor decides whether or not to stop. If auditor decides to stop, it returns a decision $Y_a \in \{\texttt{Yes}, \texttt{No}\}$, otherwise it continues to the next time step.

Furthermore, we formally define what makes a successful auditor at the end of the auditing process.

**Definition 2** (($\epsilon, \delta$)-auditor)**.** *An auditor is an ($\epsilon, \delta$)-auditor for $\epsilon, \delta \in [0, 1]$, hypothesis class $\mathcal{H}$ and a score function $s(\cdot)$ if $\forall h \in \mathcal{H}$ the following conditions hold : 1) if $s(h) > \epsilon$, $\Pr(Y_a = \texttt{Yes}) \geq 1 - \delta$ and 2) if $s(h) = 0$, $\Pr(Y_a = \texttt{No}) = 1$.*

The first condition pertaining to *soundness* implies that a successful auditor should return a $\texttt{Yes}$ with high probability when AP is followed by the model to a large extent. The second condition measures *completeness*, and requires the auditor to say $\texttt{No}$ when AP is not followed at all. Observe that when $0 < s(h) \leq \epsilon$, we cannot place a guarantee on the auditor's decision with a finite query budget.

*Query complexity $T$* of an ($\epsilon, \delta$)-auditor is the total number of queries it asks the DS before stopping. Efficient auditing requires that $T$ be small.

## 3 An Example of Auditing Through 'Feature Sensitivity'

A popular auditing property in the literature has been demographic parity. But this notion applies exclusively to fairness and is distribution dependent. In contrast, we focus on 'feature sensitivity'.

A feature is termed sensitive if changing its value in the input leads to a different prediction. Identifying such features can lead to insights into a model's working and uncover spurious correlations and harmful biases that it may have learnt. For instance, while auditing a model that predicts the presence of lung cancer based on a radiology image, a human auditor may suspect a feature in the input, such as the presence of pen markings on X-rays, to be spuriously correlated to the output. Another example is when an auditor might suspect that an input feature such as home zipcode, which might act as a proxy for race (not included in the data), may be correlated with loan rejections. In both cases, the goal is to audit if a model is sensitive to a specific input feature. Note that our presented techniques can easily extend to multiple sensitive features by repeating the same procedure for different features. When features are correlated, we assume that there is a third party, for eg. a regulator or a government entity, which guides what features are correlated and what features should be audited for.

Let the feature that we wish to audit the sensitivity of be called a feature-of-interest (FoI). Let *pair* $p_{ij}$ denote a pair of inputs $x^i, x^j \in \mathcal{X}$ which are same in all but the feature-of-interest. The pair $p_{ij}$ is called a *responsive* pair if the predictions for $x^i$ and $x^j$ are distinct. These pairs need not be constructed from a data distribution and can be artificially synthesized like in membership query active learning and therefore the correlations according to a data distribution need not be respected. Unless mentioned, predictions are made by the model $\bar{h}$. Lastly, our feature-of-interest is a sensitive feature for the model if one or more responsive pairs exist.

The score function for feature sensitivity is the probability that a randomly drawn pair from the set of all pairs $P$ is a responsive pair and is given as $s(\cdot) = \Pr_{p_{ij} \in P}(p_{ij}$ is a responsive pair$)$. When $P$ is finite, the score function can be interpreted as the fraction of responsive pairs. Usually when the score function is probabilistic, knowledge of the underlying distribution is required in order to estimate its value. However, in

the following sections, we give an auditing decision without computing the value of score function explicitly, by using the structure of the class and explanation methods. We next prove that feature sensitivity is a testable property.

**Theorem 3.1.** *Feature Sensitivity, given by $s(\cdot) = \Pr_{p_{ij} \in P}(p_{ij}$ is a responsive pair), is testable for a hypothesis class $\mathcal{H}$ using the instance space $\mathcal{X} = \mathcal{R}^d$.*

*Proof.* Fix two hypothesis $h_1, h_2 \in \mathcal{H}$ such that $\forall x \in \mathcal{X}$ it holds that $h_1(x) = h_2(x)$. Fix a pair $(x_i, x_j) \in \mathcal{X}$. This pair is responsive according to $h_1$ if and only if it is responsive according to $h_2$, i.e., $h_1(x_i) \neq h_1(x_j) \iff h_2(x_i) \neq h_2(x_j)$. Thus $s(h_1) = s(h_2)$ and it is testable for $\mathcal{H}$. □

**A Simple Baseline: Random Testing.** An easy way to detect feature sensitivity for *any* hypothesis class is to query a large number of pairs at random from the DS. The feature is sensitive if any responsive pair exists. This algorithm would need to query $O(\frac{1}{\epsilon} \log \frac{1}{\delta})$ pairs to be an $(\epsilon, \delta)$-auditor.

Note that this baseline is independent of the hypothesis class. It does not need to find out what the hidden $\bar{h}$ is, even partially, in order to audit and it doesn't use explanations either. However, a lot of pairs have to be queried for a small $\epsilon$. Next we discuss cases where using explanations and exploiting the structure of hypothesis class leads to more efficient auditing.

### 3.1 Linear Classifiers

Our first hypothesis class is *Linear Classifiers*, defined as $\mathcal{H}_{LC} = \{h_{w,b}\}_{w \in R^d, b \in R}$ where $h_{w,b}(x) = sign(\langle w, x \rangle + b)$. Responsive pairs exist with respect to $h \in \mathcal{H}_{LC}$ and for a feature of interest $x_i$ if and only if weight $w_i$ is non-zero.

#### 3.1.1 Counterfactual Explanations

Given an input $x$ and a model $h$, a counterfactual explanation Laugel et al. (2017); Deutch and Frost (2019); Karimi et al. (2020) returns the closest instance $x'$ in $L_2$ distance such that $h$ labels $x'$ differently from $x$. More precisely, $x' = \text{argmin}_{x':h(x') \neq h(x)} \|x - x'\|_2$.

We observe that for linear classifiers the difference $x - x'$ is parallel to $w$. This fact can be exploited by an auditor, which returns a decision by simply checking if $x - x'$ is zero in the feature-of-interest. We use these insights to design an auditor that uses a single random query to audit, denoted by $\texttt{AlgLC}_c$ 1.

---

**Algorithm 1** $\texttt{AlgLC}_c$ : Auditing Linear Classifiers using Counterfactuals

---

1: Query any point $x$ from the DS
2: Auditor receives label $y$ and explanation $x'$ from the DS
3: $\hat{w} \leftarrow x - x'$
4: **if** $\hat{w}_i = 0$ ($i^{th}$ feature is the FoI) **then**
5:     return No
6: **else**
7:     return Yes
8: **end if**

---

**Theorem 3.2.** *For any $\epsilon \in [0, 1]$, auditor $\texttt{AlgLC}_c$ is an ($\epsilon$,0)-auditor for feature sensitivity and hypothesis class $\mathcal{H}_{LC}$ with $T = 1$ query.*

Proof for the theorem can be found in the Appendix §A.2.

Counterfactual explanations enable the auditor to *partially* learn the hidden $\bar{h}$ via a scaled (not exact) version of the weights, which it then utilizes to make a decision. This partial-learning-based-auditing is way efficient than completely learning a linear classifier without explanations, which requires $O(d \log \frac{1}{\epsilon})$ queries in the active learning setting §4. Also note that we allow the explanation to be found from the instance

space rather than the training set. This is so because an example in the training data may be too far to be considered a good counterfactual explanation and if a counterfactual with some property (for instance an actionable counterfactual) is desired then the training data may not contain a relevant counterfactual example Karimi et al. (2020); Deutch and Frost (2019).

### 3.1.2 Anchor Explanations

Given a model $h$, an instance $x$, a distribution $D$, and a precision parameter $\tau$, an anchor explanation Ribeiro et al. (2018) returns a hyperrectangle $A_x$ such that (a) $A_x$ contains $x$ (b) at least $\tau$ fraction of the points in $A_x$ under $D$ have the same label $h(x)$ Ribeiro et al. (2018); Dasgupta et al. (2022). Specifically, $\tau = \Pr_{x' \in_D A_x}(h(x) = h(x'))$. Here, the precision parameter $\tau$ measures the quality of the anchor explanation. To describe the quality of an anchor explanation, we also use a coverage parameter, $c$ – which is the probability that a point sampled according to $D$ lies in the $A_x$, $c = \Pr_{x' \sim D}(x' \in A_x)$. The distribution $D$ is used by the DS to create anchor explanations. Refer to the appendix for notes on how to deal with imperfect precision.

We consider homogeneous linear classifiers ($b = 0$) in this section; however our techniques can be easily extended to non-homogeneous linear classifiers by considering $d + 1$ dimensions and concatenating 1 to $x$ and $b$ to $w$. For simplicity, we assume our anchor explanations have perfect precision.

We propose an auditor with anchors inspired by the active learning algorithm of Alabdulmohsin et al. (2015) which does not use explanations. Their goal is to learn a hypothesis by maintaining a search space over hypotheses and narrowing it down actively through label queries. They construct an ellipsoidal approximation of the search space at each step, $\varepsilon^\star = (\mu^\star, \Sigma^\star)$ where $\mu^\star$ is the center and $\Sigma^\star$ is the covariance matrix of the ellipsoid. Next they extract the top eigenvector of a matrix comprising of the covariance matrix of the ellipsoid. This eigenvector serves as the next synthesized query to the oracle. Our auditor uses this skeleton to audit by maintaining a similar search space and narrowing it down for $\bar{h}$.

Our contribution to the aforementioned algorithm is a new method for incorporating anchors into it – a procedure that we call *Anchor Augmentation* – for potentially higher auditing efficiency. The main idea is that anchor explanations give a region of space around a point $x$ where labels are the same as that of $x$; using this fact more synthetic *already labeled* examples can be generated for free without actually querying the DS and fed into the algorithm. Through these insights we design an auditor $\texttt{AlgLC}_a$ §2 for linear classifiers using anchors. This auditor first learns the hidden $\bar{h}$ and then checks the weight of the feature of interest to return a decision.

**Notation used in $\texttt{AlgLC}_a$** In $\texttt{AlgLC}_a$, $\varepsilon^{t\star} = (\mu^{t\star}, \Sigma^{t\star})$ denotes the largest ellipsoid that approximates the search space (corresponds to search space in our case) at time $t$ where $\mu^{t\star}$ is the center and $\Sigma^{t\star}$ is the covariance matrix of the ellipsoid at time $t$. $N^t$ is the orthonormal basis of the orthogonal complement of $\mu^{t\star}$ and $N^{t'}$ is its transpose. $\alpha^{t\star}$ is the top eigenvector of the matrix $N^{t'} \Sigma^{t\star} N^t$. In the implementation by Alabdulmohsin et al. (2015), some warm-up labeled points are supplied by the user, we denote this set as $W$. Let $W(t)$ denote the $t$-th element of this set. Let the $d^{th}$ feature be the feature of interest without loss of generality.

**Worst Case Query Complexity.** Anchor explanations returned by the DS can be absolutely consistent with the input and truthful, yet be worst-case. For instance, in the 1D case, where linear classifiers are thresholds, all the anchor points can lie on one side of the input point away from the threshold – this maintains truthfulness and consistency of the explanation – but do not help in identifying the threshold anymore than the input point itself. Generalizing this to higher dimensions, worst-case would mean that the shrinkage of the search space over hypotheses with and without explanations is the same. For the aforementioned algorithm, worst-case anchor explanations for a point $x$ is $\{\lambda x | \lambda \in \mathcal{R}^+\}$ such that all anchor points lie on a ray in the direction of point $x$.

**Lemma 1.** *Given input $x$, a worst-case anchor for $\texttt{AlgLC}_a$ is of the form $A_x = \{\lambda x | \lambda \in \mathcal{R}^+\}$ with precision parameter $\tau = 1$.*

*Proof.* Consider that given $x$, DS returns label $y$ and anchor explanation $A_x = \{\lambda x | \lambda \in \mathcal{R}^+\}$ with $\tau = 1$.

---

**Algorithm 2** $\texttt{AlgLC}_a$ : Auditing Linear Classifiers using Anchors

---

1: **Input:** $T$, augmentation size $s$, set of warm-up labeled points $W$
2: set of queried points $Q := \emptyset$, $l :=$ size$(W)$
3: **for** $t = 1, 2, 3, \ldots, T + l$ **do**
4:    **if** $t <= l$ **then**
5:       $(x, y) := W(t)$
6:       $Q \leftarrow Q \cup (x, y)$
7:       goto step 15
8:    **else**
9:       Query point $x^t := N^t \alpha^{t\star}$ from the DS
10:       Auditor receives label $y$ and explanation $A_x$ from the DS
11:       $Q \leftarrow Q \cup (x^t, y)$
12:       Sample randomly $q$ points $x^{t1} \ldots x^{tq}$ from $A_x$     Anchor
13:       $Q \leftarrow Q \cup \{(x^{t1}, y) \ldots (x^{tq}, y)\}$     Augmentation
14:    **end if**
15:    $\varepsilon^{(t+1)*} = \left(\mu^{(t+1)\star}, \Sigma^{(t+1)\star}\right) :=$ estimate_ellipsoid$(Q)$
16:    $N^{t+1} :=$ update_N$(\mu^{(t+1)\star})$
17:    $\alpha^{(t+1)\star} :=$ update_alpha$(N^{t+1}, \Sigma^{(t+1)\star})$
18: **end for**
19: $\hat{w} = \mu^{T+1}$
20: **if** $|\hat{w}_i| \leq \Delta$ ($i^{th}$ feature is the FoI) **then**
21:    return $\texttt{No}$
22: **else**
23:    return $\texttt{Yes}$
24: **end if**

---

Let the set of all classifiers consistent with the label be $H = \{w : y\langle w, x\rangle > 0\}$.

From the definition of anchors and $\tau = 1$, the label of all points in anchor $A_x$ is also $y$. Then, the set of all classifiers consistent with $A_x$ is, $H' = \{w : y\langle w, \lambda x\rangle > 0\} = \{w : \lambda y\langle w, x\rangle > 0\} = H$. Hence the set of consistent classifiers remains the same despite anchors. Therefore $A_x$ qualifies as a worst-case anchor. $\quad\square$

The query complexity of our auditor using anchors $\texttt{AlgLC}_a$ is presented below.

**Theorem 3.3.** *For every dimension d, there exists $C > 0$ such that for any $\epsilon \in (0, 1)$, auditor $\texttt{AlgLC}_a$ is an $(\epsilon, 0)$-auditor for feature sensitivity and $\mathcal{H}_{LC}$ with $T = O\left(d \log \frac{2C}{\epsilon}\right)$ queries.*

This bound is comparable to actively learning a linear classifier without explanations $\mathrm{O}(d \log \frac{1}{\epsilon})$ Alabdulmohsin et al. (2015); Balcan and Long (2013) and might discredit the utility of explanations. However, we show the silver lining in §5.1. Proof and more details about the algorithm can be found in Appendix §A.4.

## 3.2 Decision Trees

Now we move beyond the linear hypothesis class to a non-linear one – decision trees and show how explanations can help in auditing them. A natural explanation for a decision tree prediction is the path traversed in the tree from root to the predicted leaf Audemard et al. (2021); Boer et al. (2020); Dasgupta et al. (2022). This explanation is consistent if the explanation path leads to the same prediction as that returned by the data scientist and the nodes with their values/range are consistent with the input features.

We propose an auditor $\texttt{AlgDT}$ based on an explanation-based breadth-first tree search algorithm, it works as follows. If the feature-of-interest is not present in the explanation path, auditor randomly picks a node in the path and perturbs its value while keeping the other features fixed such that it can explore the other branch of the perturbed node. Incase after perturbation the point satisfies one of the previous paths received as an

explanation, it randomly picks a new query. Otherwise, the feature-of-interest is a node in the explanation, and therefore is a sensitive feature since it is used in the tree. `AlgDT` can be found in 3.

---

**Algorithm 3** `AlgDT` : Auditing Decision Trees using decision path explanations

---

1: Given : Query Count threshold $qT \geq 1$
2: Initialize : Query count $qcount := 0$; set of received paths, $P := \emptyset$
3: Query any point $x$ from the DS
4: Auditor receives label $y$ and explanation path $p$ from the DS
5: $P \leftarrow P \cup p$
6: $qcount := qcount + 1$
7: **if** $foi$ is a node in $p$ **then**
8:     return `Yes`
9: **end if**
10: **while** True **do**
11:     **if** $qcount == qT$ **then**
12:         return `No`
13:     **end if**
14:     $x := \mathrm{perturb}(x, p)$ {In App. Sec.A.5 }
15:     **while** $\mathrm{findpath}(x, P)$ {In App. Sec.A.5 } **do**
16:         Pick a random query as $x$
17:     **end while**
18:     Query $x$ from the DS
19:     Auditor receives label $y'$ and explanation path $p'$ from the DS
20:     $P \leftarrow P \cup p'$
21:     $qcount := qcount + 1$
22:     **if** $foi$ is a node in $p'$ **then**
23:         return `Yes`
24:     **end if**
25: **end while**

---

While we are not aware of an active learning algorithm that learns a decision tree with continuous features, Kushilevitz and Mansour (1991) propose a non-explanation membership query algorithm for binary features that exactly learns the tree in time $\mathrm{poly}(2^{\mathrm{depth}}, \mathrm{d})$. Our auditor with explanations, which basically learns the entire tree in the worst-case, has a linear worst-case complexity on the order of number of nodes in the tree. Proof is in the App. Sec. A.5.

**Theorem 3.4.** *For any $\epsilon \in [0, 1]$, auditor `AlgDT` is an ($\epsilon$,0)-auditor for feature sensitivity and hypothesis class $\mathcal{H}_{DT}$ with $T = O(V)$ queries where $V$ is the number of nodes in the decision tree.*

## 4 Auditor through connections with Active Learning

In the previous section we designed specific auditors based on hypothesis class and explanation type. This naturally begs the question if a general auditing strategy exists, we answer in the affirmative. We start by taking a closer look at two critical steps – strategy to pick the next query and the stopping condition – of our interactive auditing protocol mentioned in §2 and then propose a general recipe to create an auditor.

**Picking next query** The next query should be picked such that the auditor gets closer to the hidden $\bar{h}$, thereby reducing its uncertainty in making a decision. To achieve this the auditor can maintain a search space over hypotheses beginning with $\mathcal{S}_0 = \mathcal{H}$ and narrowing it through queries made to the DS at each subsequent step until the stopping condition is reached.

Table 1: Worst-Case Query Complexity of auditing feature sensitivity

| Auditor | Query Complexity |
|---------|------------------|
| Baseline | $O(\frac{1}{\epsilon} \log \frac{1}{\delta})$ |
| `AlgLC`$_c$ | 1 |
| `AlgLC`$_a$ | $O(d \log(\frac{2c}{\epsilon}))$ |
| `AlgDT` | $O(V)$ |

Let $Z_{t-1} \subseteq Z = \mathcal{X} \times \mathcal{Y}$ and $E_{t-1} \subseteq \mathcal{X} \times \mathcal{E}$ be the set of labeled examples and explanations acquired from the DS till $t-1$. Search space at $t$, $\mathcal{S}_t$, is the subset of hypotheses that are consistent with all the labels *and explanations* in $Z_{t-1}$ and $E_{t-1}$ respectively. As an example for counterfactual explanations, the hypothesis $h$ is consistent if $\forall (x, \bar{h}(x)) \in Z_{t-1}$ and their counterfactuals $(x, x' = e_{\bar{h}}(x)) \in E_{t-1}$, $h(x) = \bar{h}(x)$ and $h(x') = \bar{h}(x')$.

For efficient pruning of the search space, the auditor chooses a query point that reduces the measure of the search space the most, no matter what explanation and label are returned by the DS. Formally, $x_t = \mathrm{argmax}_{x \in \mathcal{X}}(\min_{\mathcal{E} \times \mathcal{Y}} |\mathcal{S}_t|/|\mathcal{S}_{t+1}|) = \mathrm{argmax}_{x \in \mathcal{X}} \text{ value}_x$ where $|\mathcal{S}_t|$ and $|\mathcal{S}_{t+1}|$ denote the measure of the search spaces $\mathcal{S}_t$ and $\mathcal{S}_{t+1}$ and $\text{value}_x$ corresponds to the reduction by worst case label and explanation for a point $x$. For finite hypothesis classes, the measure of the search space corresponds to the number of elements in it, while for infinite classes it is more complex; in §3.1.2 the search space is approximated by an ellipsoid at each time and measure is volume of the ellipsoid.

**Stopping Condition**   decides query complexity of the auditor. For finite hypothesis classes, since $\bar{h}$ is always guaranteed to be in the search space at all times, if $s(h) > \epsilon$ for all $h$ in $S_t$ then the auditor can safely conclude that $\bar{h}$ has the desired property and stop with a `Yes` decision. Similarly, if $s(h) \leq \epsilon$ for all $h$ in $S_t$ the auditor will stop with a `No`. Observe that the auditor will ultimately arrive at one of these two cases – if none of the two stopping conditions hold, the next query will cause the search space to shrink – and ultimately our search space will consist of a single hypothesis.

For *infinite* hypothesis classes, we propose exploiting the structure of the hypothesis class and explanation methods to reduce the search space to a single hypothesis or to a set where value of the score function is same for each of its elements, in which case similar stopping conditions as the finite case can be used. Our proposed auditors for feature sensitivity lie in this setting.

Using the above query picking and stopping strategies, we formally outline an auditor for finite hypothesis classes presented in Alg.4 in the Appendix Sec.A.1. These strategies lead to an $(\epsilon, \delta)$-auditor, which follows directly from $\bar{h}$ belonging to the search space at all times and the stopping conditions as proved below.

**Theorem 4.1.** *Algorithm 4 is an $(\epsilon, \delta)$-auditor for a finite hypothesis class $\mathcal{H}$.*

*Proof.* Firstly, note that $\bar{h}$ is in the search space $S_t$ at all times $t$. This is because the search space is reduced based on labels and explanations w.r.t. $\bar{h}$ (provided by the data scientist). Next, we assume that value of function $s$ can be computed to arbitrary precision for any hypothesis $h$. Then the proposed stopping conditions satisfy the soundness and completeness properties of the $(\epsilon, \delta)$-auditor from definition. $\square$

This general auditor is similar to the Optimal Deterministic Algorithm of Yan and Zhang (2022) in the sense that it is inspired from active learning. However, our general auditor has different stopping condition, also uses explanations and is not manipulation-proof (although all of our auditors in §3 are manipulation-proof). Due to difference in stopping condition and the fact that our auditor returns a binary decision, our auditor might be faster in reality. See Appendix §A.6 for details on manipulation-proofness.

**Connections to Active Learning.** For readers who are acquainted with active learning, our auditor may appear familiar. In active learning Dasgupta (2005), the goal is to learn a classifier in an interactive manner by querying highly informative unlabeled data points for labels. A specific variant of active learning is Membership Query Active Learning (MQAL) Angluin (1988); Feldman (2009) where the learner synthesizes queries rather than sampling from the data distribution or selecting from a pool.

Our auditor is essentially an active *partial*-learning algorithm in the MQAL setting, with a *modified* oracle that returns explanations in addition to labels. Here, the DS is the oracle and the auditor is the learner. The algorithm does *partial* learning – since it stops when the auditing goal is complete and before the classifier $\bar{h}$ is fully learnt.

We observe that learning is a harder task than auditing – since the goal in learning is to find the classifier generating the labels, while in auditing we aim to decide if this classifier has a certain property. This implies that if there exists an algorithm that can learn a classifier, it can also audit it as long as the score function

can be computed/estimated, and a membership query active learning algorithm can be used as a fallback auditing algorithm. This leads to the following fallback guarantee.

**Theorem 4.2.** *If there exists a membership query active learner that can learn $\bar{h}$ exactly in $T$ queries, then the active learner can also audit in $T$ queries.*

*Proof.* Once $\bar{h}$ is known, $s(\bar{h})$ returns the auditing decision. If $s(\bar{h}) = 0$, the decision is No and if $s(\bar{h}) > \epsilon$, the decision is Yes. $\qquad\square$

To summarize, while our auditor is connected to active learning, the key differences are – 1) the goals as mentioned above, 2) usage of explanations and 3) unlike active learning, partial learning of the hypothesis can be sufficient for auditing. For example, if all hypotheses in the search space have a zero score value, auditor can return a decision without learning $\bar{h}$ exactly; or learning only some parameters of $\bar{h}$ may be enough to audit. Since partial learning can be sufficient for auditing, an auditing algorithm is not necessarily an active learning algorithm.

**Distribution-Dependent Properties.** For distribution-dependent properties, the score function should also depend on the distribution and hence the auditor will have to estimate the score function using samples from this distribution (or another distribution which will lead to errors in the estimate based on how different the two distributions are).

The general idea to audit a distribution-dependent property such as demographic parity under our framework is to find the hidden classifier through active learning queries and estimate the score function for the learnt classifier by sampling from the said distribution. For finite hypothesis class, it can also use algorithm 4 except that the score function for all existing hypotheses in the search space is to be estimated using samples from the said distribution. Note that while these algorithms are manipulation-proof unlike using random sampling for estimating the property as noted in Yan and Zhang (2022).

## 5 Experiments

In this section we conduct experiments on standard datasets to test some aspects of feature sensitivity auditing. Specifically, we ask the following questions:

1. Does augmentation of 'typical' anchors reduce query complexity for linear classifiers?
2. How many queries are needed on average to audit decision trees of various depths?

The datasets used in our experiments are - Adult Income Becker and Kohavi (1996), CovertypeBlackard (1998) and Credit DefaultYeh (2016). The features of interest are gender, wilderness area type and sex for Adult, Covertype and Credit datasets respectively. All the datasets have a mix of categorical and continuous features. Categorical features are processed such that each category corresponds to a binary feature in itself. We remove all rows with missing values in any columns. We use an 80-20 split to create the train-test sets.

We learn both our linear and tree classifiers using scikit-learn. To run the anchor experiments, we use the matlab code provided by Alabdulmohsin et al. (2015). All of our experiments on a CPU varying from a minute to 2-3 hours. Additional details about the experiments can be found in the appendix §A.7.

### 5.1 Anchor Augmentation of Typical Anchors

As discussed in §3.1.2, augmentation of worst-case anchors does not help in reducing the query complexity of auditing. In other words, augmenting worst-case anchors is equivalent to not using anchor explanations. A natural question then is - do typical anchors help? Since auditing with anchors using $\texttt{AlgLC}_a$ means learning the DS's model under the hood, reduction in query complexity of learning implies reduction in that of auditing. Hence, we check experimentally if faster learning is achieved through anchor augmentation of typical anchor points.

**Methodology** We learn a linear classifier with weights $w$ for each dataset ; these correspond to the DS's model. All of these models have a non-zero weight for the feature of interest; later we manually set the

weight corresponding to the FoI to zero to conduct experiments for the case when the FoI is not used for auditing. Then the weights are estimated during auditing using Algorithm 2. We consider two different augmentations 1) worst-case anchor points which is equivalent to not using anchors 2) typical anchor points. We set the augmentation size (number of anchor points augmented) to a maximum of 30. Our anchors are hyperrectangles of a fixed volume surrounding the query point. We sample points with the same label as the query point from this hyperrectangle and augment to the set of queries in the typical case.

**Results** The results for the non-zero FoI case are shown in Figure 1. We see that the anchor augmentation of typical anchors drastically reduces the query complexity to achieve the same estimation error between weights as compared to not using anchors (or equivalently using worst-case anchors). This saving directly translates to efficiency in auditing. For example, in the adults dataset, less than 50% of the queries are needed to achieve a lower error with typical anchors than without them. This illustrates that anchor explanations can be helpful for auditing, suggesting an application for these explanations. Note that the error in Figure 1 is not zero and could be due to the ellipsoidal approximation. When the FoI has a weight of zero, we observe that typical anchors reach a stable near-zero estimate for the FoI weight sooner than worst-case anchors, as shown in Figure 2.

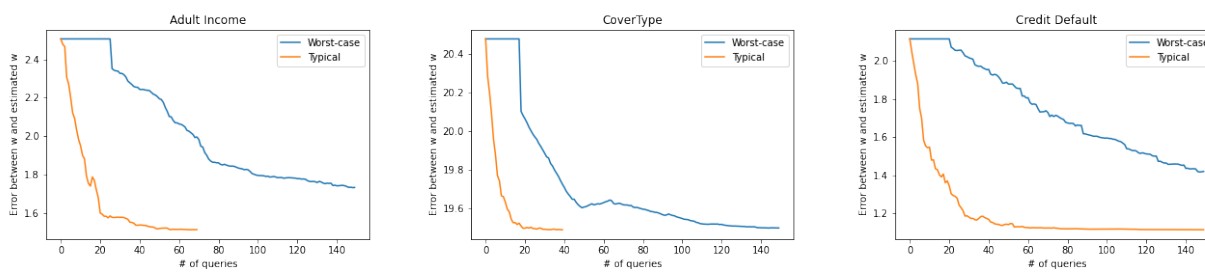

Figure 1: Number of queries required to learn a linear classifier with anchor augmentation of worst-case anchor points (in blue) and typical anchor points (in orange). For the latter, a clear reduction in query complexity is observed. The FoI has a non-zero weight in all of these.

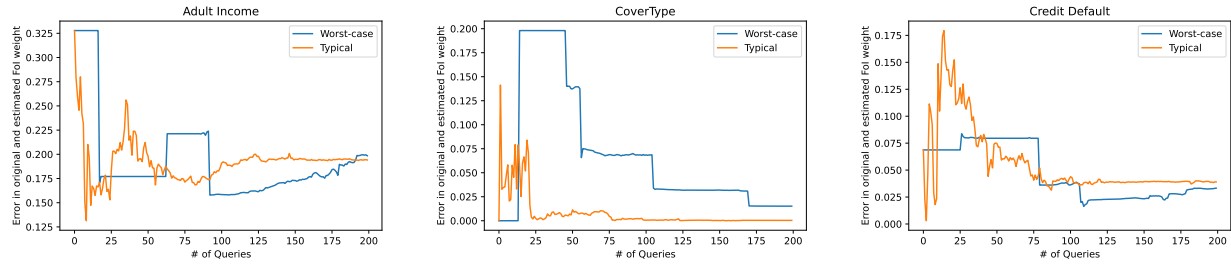

Figure 2: Number of queries required to learn a linear classifier when the FoI has a zero weight, with anchor augmentation of worst-case anchor points (in blue) and typical anchor points (in orange). The latter reaches a stable near-zero weight for the FoI sooner, demonstrating a reduction in query complexity.

## 5.2 Average Queries for Decision Tree Auditing

In §3.2 we discussed that the worst-case query complexity of auditing is on the order of number of nodes in the decision tree. However on average the auditor might ask way lesser queries than the worst-case bound. We check experimentally if this is the case.

**Methodology** We learn decision trees for each of the datasets using scikit-learn which implements CART Breiman (2017) to construct the tree. All of these use the FoI to make predictions. We vary tree depth by fixing the 'max-depth' hyperparameter. Then we freeze the tree and run `AlgDT` a 1000 times. We report an

average of the total queries required to audit across the 1000 runs. We also run our random testing baseline from §3 until we detect the sensitive feature. We do this a 1000 times and report the average number of queries over those runs.

**Results** Our results are displayed in Table 2. As can be seen, the average number of queries needed to audit is consistently way lesser than the number of nodes, which highlights the utility of explanations in the average-case. There is a complex relationship between the total number of nodes in the tree and the number of feature-of-interest nodes in the tree which jointly determines the number of queries.

As the depth of the tree increases, the number of nodes increase exponentially while the feature-of-interest nodes and therefore average queries increase at a much slower pace. The CovType dataset has the lowest query complexity which might be a result of the feature-of-interest having four categories rather than two (male/female) in other datasets. There is a huge disparity between the average number of queries our auditor needs (low) and the average random testing queries (high). This demonstrates the efficacy of our auditor over the baseline.

Table 2: Decision Tree Auditing : Avg. number of queries required to audit over 1000 runs of `AlgDT` across different depths of decision trees. #Avg. queries are rounded to the nearest integer and reported with 95% confidence intervals. #Avg. queries are consistently lower than #Nodes and #Random Testing Queries.

| Dataset | Depth | Test Acc. | #Avg. Queries | #Nodes | #FoI Nodes | #Random Testing Queries |
|---------|-------|-----------|---------------|--------|------------|--------------------------|
| Adult   | 9     | 85.09     | $11 \pm 0.48$ | 188    | 3          | $248 \pm 15.4$           |
|         | 12    | 85.36     | $9 \pm 0.4$   | 552    | 18         | $299 \pm 18.65$          |
|         | 15    | 84.59     | $10 \pm 0.41$ | 1158   | 31         | $160 \pm 9.8$            |
| CovType | 7     | 77.92     | $2 \pm 0.10$  | 119    | 9          | $17 \pm 1.03$            |
|         | 9     | 80.06     | $2 \pm 0.07$  | 383    | 14         | $13 \pm 0.77$            |
|         | 12    | 83.80     | $2 \pm 0.08$  | 1625   | 34         | $11 \pm 0.67$            |
|         | 15    | 87.40     | $2 \pm 0.08$  | 4196   | 57         | $9 \pm 0.5$              |
| Credit  | 9     | 81.07     | $17 \pm 0.63$ | 234    | 3          | $176 \pm 10.71$          |
|         | 12    | 80.58     | $36 \pm 1.2$  | 629    | 6          | $276 \pm 17.59$          |
|         | 15    | 81.65     | $54 \pm 1.87$ | 1030   | 20         | $905 \pm 54.31$          |

To summarize, we observe that worst-case estimates are generally very pessimistic and project explanations in a bad light. In the average case, which happens a lot on average, explanations bring down the number of queries for auditing significantly.

## 6 Discussion

**Untruthful Data Scientist** A natural question to ask is what happens when a DS is not entirely truthful in the auditing process, as we assumed in §2. What kind of auditing is possible in this case?

Suppose the DS returns all labels and explanations from an entirely different model than $\bar{h}$. In this case, there is no way for an auditor to detect it; but perhaps this can be dealt with in a procedural manner – like the DS hands its model to a trusted third party who answers the auditor's queries.

What if the DS returns the correct labels but incorrect explanations? DS might be forced to return correct labels when the auditor has some labeled samples already and hence can catch the DS if it lies with labels. This scenario motivates verification of explanations. Next, we discuss schemes to verify anchors and counterfactuals for this scenario.

**Anchors.** For anchors, verification is possible if we have samples from the underlying distribution $D$. For a query $x$, DS returns an unverified anchor $A_x$ with precision $\tau_x$ and coverage $c_x$. The correctness of $\tau_x$ can be detected as follows. First, get an estimate for the true value of the precision parameter by sampling $n$ points from anchor $A_x$ according to $D$. Let the true and estimated values of precision parameter be $\tau$ and $\hat{\tau}$ respectively. $\hat{\tau}$ approaches $\tau$ with sufficiently large number of points as mentioned in lemma 2. Second, compare $\tau_x$ with $\hat{\tau}$. A large difference between $\tau_x$ and $\hat{\tau}$ implies false anchors.

**Lemma 2.** *For any $\Delta > 0$ and integer $n$, $\Pr(|\tau - \hat{\tau}| \geq \Delta) \leq 2\exp^{-2\Delta^2 n}$.*

Lemma 2 is immediate from Hoeffding's Inequality. Notice that the number of samples $n$ required to verify precision changes with $1/\Delta^2$. If the fraction of responsive pairs $\epsilon$ equals $\Delta$, our baseline in section 3 can audit with $O(1/\Delta)$ samples without using explanations. Hence, in adversarial conditions where the probability of a lying DS is high, it is better to audit with our explanations-free baseline than auditing with anchors and verifying them.

Since coverage is the probability that a point sampled from $D$ belongs to $A_x$, it can be easily checked 1) by calculating the volume of $A_x$ when all dimensions of $x$ are bounded or 2) by sampling points from $D$ when the features are unbounded.

**Counterfactuals.** Given $x$, let $x'$ be the unverified counterfactual explanation returned by the DS. There are two aspects to a counterfactual explanation – its label which should be different from $x$ and it should be the closest such point to $x$.

The first aspect can be easily verified by querying $x'$ from the DS. For the second aspect, we observe that finding the counterfactual is equivalent to finding the closest adversarial point. Deriving from adversarial robustness literature, verifying the closeness aspect can be a computationally hard problem for some hypothesis classes like discussed in Weng et al. (2018). However, we propose a sampling based algorithm to estimate the true counterfactual, assuming that the DS is lying. Firstly, sample points from the ball $B(x, d(x, x'))$. For $x'$ to be the true counterfactual, all points within the ball $B(x, d(x, x'))$ should have the same label as $x$. If a point $x''$ with a different label is sampled, select this point as an estimate of the true counterfactual and repeat the scheme with the new ball $B(x, d(x, x''))$. By following this procedure iteratively, we get closer to the correct counterfactual explanation as the radius of the ball reduces at each iteration. There are cases where our algorithm may not work well. We leave designing better algorithms for verifying closeness to future work.

Upon verification, if auditor finds that the DS is untruthful, it can choose to 1) stop auditing and declare that the DS is lying, 2) audit with estimated explanations or 3) audit with explanations-free baseline algorithm (§3) since option 2 is computationally intensive.

**Privacy** Auditing also raises a legitimate privacy concern. On one hand, model is hidden from the auditor for confidentiality reasons while on the other hand, to be able to give a correct auditing decision efficiently, the auditor has to extract/learn the hidden model, albeit partially. This concern can be resolved by using crytographic tools like Zero-Knowledge proofs Yadav et al. (2024); Singh et al. (2021); Liu et al. (2021). A future direction of our work is modifying our framework to work with cryptographic tools.

## 7 Related Work

Auditing has been used to uncover undesirable behavior in ML models in the past Buolamwini and Gebru (2018); Koenecke et al. (2020); Tatman and Kasten (2017); Tatman (2017). However, these works were based on creating a audit dataset in an ad-hoc manner. Recently, there have been efforts towards streamlining and developing a structured process for auditing. For instance, Gebru et al. (2018); Mitchell et al. (2019) introduce datasheets for datasets and model cards and Raji et al. (2020) introduce an end-to-end internal auditing process. In contrast with these works, we theoretically formalize auditing with explanations which allows us to determine what kind of properties can be audited and with how many queries.

**Auditing with Formal Guarantees** The work most related to ours is concurrent work due to Yan and Zhang (2022), who present a formal framework as well as algorithms for auditing fairness in ML models by checking if a model has a certain demographic parity on a data distribution. Their algorithms are motivated by connections to active learning as well as machine teaching Goldman and Kearns (1995). In contrast, we look at auditing *with explanations*. Our instantiation – auditing feature sensitivity – is different from demographic parity, which, together with the use of explanations, leads to very different algorithms.

Another work on auditing with guarantees is Goldwasser et al. (2021), who audit the accuracy of ML models. They use an interactive protocol between the verifier and the prover like our work, and their algorithm is related to a notion of property testing due to Balcan et al. (2011). The auditing task used in our paper, feature sensitivity, is different from accuracy used in their paper. We also utilize explanations.

**Auditing and Explanations** Explanations are supposed to help in auditing as suggested in Bhatt et al. (2020); Oala et al. (2020); Adebayo and Gorelick (2017); Watson and Floridi (2021); Zhang et al. (2022); Poland (2022); Hamelers (2021); Carvalho et al. (2019), but exactly how and under what conditions was not understood. In a concurrent work by Mougan et al. (2023), authors show that explanations lead to more sensitive audits than using labels and demonstrate the case for demographic parity using shapley values Lundberg et al. (2019; 2020). On the other hand, our focus is to demonstrate how explanations can reduce the query complexity of auditing and formalizing the auditing process by connecting it to active learning. The notion of feature sensitivity and the kind of explanations used in our paper are also different from theirs.

**Other Auditing Methods** Specific statistical methods have been proposed by Jagielski et al. (2020); Ye et al. (2021) to audit privacy and by Liu and Tsaftaris (2020); Huang et al. (2021) to audit data deletion. These are properties of the *learning algorithm* rather than the *model*. In our paper we focus on the latter and hence our algorithms are not applicable to auditing privacy and data deletion. Extending our framework to broader settings is an important direction of future work.

## 8 Conclusions and Future Work

We formalize the role of explanations in auditing with a special focus on 'feature sensitivity' and find that their theoretical efficacy in reducing query complexity vastly varies across different methods. Structure of the hypothesis class and explanation method together determine this efficacy. While worst-case query complexities for auditing 'feature sensitivity' are pessimistic, our experiments herald optimism – in the average-case we see a significant reduction in the number of queries required to audit across all explanation types. We also connect auditing to the well-established field of active learning and provide general active-learning inspired algorithms for auditing with explanations.

We believe that this work is a first step towards understanding certified auditing and auditing with explanations. A major research direction is auditing with approximate or incorrect explanations. Other future directions include incorporating other kinds of explanations, investigating different hypothesis classes, auditing properties and theorising the average-case complexity with explanations.

**Ethics Statement.** This work uses feature sensitivity more broadly than fairness (as described in §3)and only to demonstrate an instantiation of our framework. We agree that feature sensitivity, though linked to fairness, is not 'the' perfect measure of fairness and we doubt if such a metric can exist.

We believe that auditing is a complex process. Auditing can only be done for vulnerabilities that we know can exist and hence unknown vulnerabilities might still exist in a model despite being audited. There are real-life scenarios like limited budget (financial, time) which might prevent an auditor from *uncovering* vulnerabilites, even though they exist. Our work is a step in understanding the auditing process better.

**Acknowledgements.** This work was supported by NSF under CNS 1804829 and ARO MURI W911NF2110317. MM has received funding from the European Research Council (ERC) under the European Union's Horizon 2020 research and innovation program (grant agreement No. 882396), by the Israel Science Foundation (grant number 993/17), Tel Aviv University Center for AI and Data Science (TAD), and the Yandex Initiative for Machine Learning at Tel Aviv University. CY thanks Geelon So for providing valuable feedback on an early draft of the paper.

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

# A   Appendix

## A.1   A General Auditor for finite hypothesis classes

---
**Algorithm 4** A General Auditor for finite hypothesis classes
---
1: $\mathcal{S}_0 := \mathcal{H}$ where $|\mathcal{H}|$ is finite;   stop_flag := False;   $t := 1; \epsilon := eps$
2: **while** !stop_flag **do**
3:     $x_t := \text{picking\_next\_query}(\mathcal{S}_{t-1})$
4:     Auditor receives label $y_t$ and explanation $e_t$ from the DS
5:     $\mathcal{S}_{t+1} := \text{update\_search\_space}(\mathcal{S}_t, x_t, y_t, e_t)$
6:     Y_a, stop_flag = check_stopping_condition$(\mathcal{S}_t)$
7: **end while**
8: return Y_a
---

---
**Algorithm 5** check_stopping_condition()
---
1: **Input:** $\mathcal{S}_t$
2: **if** $\forall h \in \mathcal{S}_t, s(h) > eps$ **then**
3:     decision = Yes, stop_flag = True
4: **else if** $\forall h \in \mathcal{S}_t, s(h) \leq eps$ **then**
5:     decision = No, stop_flag = True
6: **else**
7:     decision = None, stop_flag = False
8: **end if**
9: return decision, !stop_flag
---

---
**Algorithm 6** picking_next_query()
---
1: **Input:** $\mathcal{S}_t$
2: **for** $x \in \mathcal{X}$ **do**
3:     **for** $(e, y) \in \mathcal{E} \times \mathcal{Y}$ **do**
4:         $\mathcal{S}_{t+1} = \text{update\_search\_space}(\mathcal{S}_t, x, y, e)$
5:     **end for**
6:     $\text{value}_x = \min_{\mathcal{E} \times \mathcal{Y}} |\mathcal{S}_t|/|\mathcal{S}_{t+1}|$
7: **end for**
8: return $\text{argmax}_{x \in \mathcal{X}} \text{value}_x$
---

---
**Algorithm 7** update_search_space()
---
1: **Input:** $\mathcal{S}, x, y, e$
2: $\mathcal{S}_{new} := \{h \in \mathcal{S} | h(x) = y, \text{check\_consistent\_explanation}(x, e_h(x), e)\}$ {//Explanation method dependent consistency check}
3: return $\mathcal{S}_{new}$
---

## A.2   Auditing Linear Classifiers with Counterfactual Explanations

In this section, we will prove that auditing linear classifiers using counterfactual explanations requires only one query. We denote our auditor by $\texttt{AlgLC}_c$, as outlined in Alg. 1. The proof goes by noting that the counterfactual explanation $x'$ returned by the DS is very close to the projection of input $x$ and that $x - x'$ is parallel to $w$. We consider the $d$-th feature to be our feature of interest without loss of generality.

**Lemma 3.** *Given hyperplane $w^T x + b = 0$, point $x$ and its projection on the hyperplane $x''$, $x - x'' = \lambda w$ where $\lambda = \frac{w^T x + b}{\|w\|_2^2}$.*

---

**Algorithm 8** $\texttt{AlgLC}_c$ : Auditing Linear Classifiers using Counterfactuals

---

1: Query any point $x$ from the DS
2: Auditor receives label $y$ and explanation $x'$ from the DS
3: $\hat{w} \leftarrow x - x'$
4: **if** $\hat{w}_i = 0$ ($i^{th}$ feature is the FoI) **then**
5:     return No
6: **else**
7:     return Yes
8: **end if**

---

*Proof.* The projection, $x''$, of $x$ on the hyperplane $w^T x + b = 0$, is found by solving the following optimization problem.

$$
\begin{aligned}
\min_{x''} \quad & \|x'' - x\|^2 \\
\text{s.t.} \quad & w^T x'' + b = 0
\end{aligned}
\tag{1}
$$

Let $L(x'', \lambda)$ be the lagrangian for the above optimization problem.

$$
\begin{aligned}
L(x'', \lambda) &= \|x'' - x\|^2 + 2\lambda(w^T x'' + b) \\
&= \|x''\|^2 + \|x\|^2 - 2x^\top x'' + 2\lambda w^\top x'' + 2\lambda b \\
&= \|x''\|^2 + \|x\|^2 - 2(x - \lambda w)^\top x'' + 2\lambda b
\end{aligned}
\tag{2}
$$

Taking derivative of the lagrangian with respect to $x''$ and equating with zero we get,

$$
\begin{aligned}
\frac{\partial L}{\partial x''} &= 2x'' - 2(x - \lambda w) = 0 \\
x - x'' &= \lambda w
\end{aligned}
\tag{3}
$$

By substituting above equation in the constraint for the optimization problem $w^T x'' + b = 0$, we get $\lambda = \frac{w^\top x + b}{\|w\|_2^2}$.

$\square$

**Theorem 3.2.** *For any $\epsilon \in [0, 1]$, auditor $\texttt{AlgLC}_c$ is an ($\epsilon$,0)-auditor for feature sensitivity and hypothesis class $\mathcal{H}_{LC}$ with $T = 1$ query.*

*Proof.* Recall that the counterfactual explanation returned by the DS for input $x$ is given as $x' = \arg\min_{x':h(x') \neq h(x)} d(x, x')$ where $d(x, x') = \|x - x'\|_2$.

The projection of $x$ on the hyperplane, $x''$ is the closest point to $x$ on the hyperplane. Therefore $x' = x'' + \Delta$ where $\Delta$ is a vector in the direction of $w$, $\Delta = \gamma w$, $\gamma$ is a very small non-zero constant.

Therefore, $x - x' = x - (x'' + \Delta) = (x - x'') + \Delta$.
Using lemma 3,

$$
\hat{w} := x - x' = \lambda w + \gamma w = c_0 w,
\tag{4}
$$

where $c_0$ is a non-zero constant. ($x - x'$ is non-zero due to the definition of counterfactuals, specifically that they have different labels.)

If $w_d = 0$, then it implies that $\hat{w}_d = 0$ and the feature has no effect on the prediction. Thus the score function is zero. Since $\texttt{AlgLC}_c$ returns a No when $\hat{w}_d = 0$, it is always correct in this case. For all the other cases when $w_d \neq 0$, it implies that $\hat{w}_d \neq 0$ and therefore, the feature has an effect on the prediction. Since $\texttt{AlgLC}_c$ returns a Yes when $\hat{w}_d \neq 0$, it is always correct.

Also $\delta = 0$ since our auditor and DS are deterministic.

$\square$

Note that this is partial learning since 1) we do not need to learn $w$ exactly and 2) we do not need to learn the bias term $b$.

### A.3   Connection between Model Parameters and Score Function

**Notation** For vector $v$, the $i^{th}$ feature is denoted $v_i$.
Let hypothesis $h_{w,b} \in \mathcal{H}_{LC}$. When $w, b$ are clear from the context, we simply write $h$. Let $w'$ be the $(d-1)$-dimensional vector $[w_1 \ldots w_{d-1}]^T$. Hence $w$ is a concatenation of $w'$ and $w_d$, denoted by the shorthand $w = [w', w_d]$. We assume that $\|w'\|_2 = 1$.
Let $x$ be a $d$-dimensional input to this hypothesis. Let $x'$ be the $(d-1)$-dimensional vector $[x_1 \ldots x_{d-1}]^T$. Let $\bar{x} = [x_1 \ldots x_{d-1}, 1]^\top$. Without loss of generality, let the $d^{th}$ feature be the feature of interest and $x_d \in \{0,1\}$.
The score function for a hypothesis $h$ is given as, $s(h) = \Pr\left((x^i, x^j) \text{ forms a responsive pair}\right)$ where $(x^i, x^j)$ is sampled uniformly from the set of all pairs and labeled by $h$. Henceforth we use this score function.

In the following theorem, we bound the fraction of responsive pairs, also our score function, using the weight of our feature of interest, $w_d$. The score function is bounded by $c \cdot |w_d|$ where $c$ depends on the dimension of the input. This implies that if $|w_d|$ is small, there are not a lot of responsive pairs (low score function value).

**Theorem 1.** *Assume $\forall x \in \mathcal{X}, \|\bar{x}\|_2 \leq 1$. Let $h_{[w', w_d], b} \in \mathcal{H}_{LC}$. Then*

$$s(h) \leq c \cdot |w_d| \tag{5}$$

*where $c = \dfrac{2^{d-2}}{\pi^{\left(\frac{d-1}{2}\right)} \cdot \Gamma\left(\frac{d+1}{2}\right)}$ is a constant for finite dimension $d$ and $\Gamma$ is Euler's Gamma function.*

*Proof.* Let $P$ be the set of all pairs of points. Let $x^i$ and $x^j$ denote two inputs forming a pair. Let the pair $(x^i, x^j)$ drawn uniformly from $P$ form a responsive pair.

From the definition of a pair, $x^i$ and $x^j$ only differ in the $d^{th}$ feature. Hence,

$$
\begin{aligned}
w^\top x^j + b &= w'^\top x'^j + b + w_d x_d^j \\
&= w'^\top x'^i + b + w_d \left(1 - x_d^i\right) \\
&= w'^\top x'^i + b + w_d - w_d x_d^i.
\end{aligned}
\tag{6}
$$

Without loss of generality, let $x_d^i = 0$, hence $x_d^j = 1$.
Therefore,

$$w^\top x^j + b = w'^\top x'^i + b + w_d \tag{7}$$

Next, writing the definition of a responsive pair for $\mathcal{H}_{LC}$ we get,

$$\text{sign}\left(w^\top x^i + b\right) \neq \text{sign}\left(w^\top x^j + b\right) \tag{8}$$

Substituting eq. 7 into the RHS of eq. 8, we get,

$$\text{sign}\left(w^\top x^i + b\right) \neq \text{sign}\left(w'^\top x'^i + b + w_d\right) \tag{9}$$

Expanding the LHS of eq. 9 and substituting $x_d^i = 0$, we get,

$$\text{sign}\left(w'^\top x'^i + b\right) \neq \text{sign}\left(w'^\top x'^i + b + w_d\right) \tag{10}$$

Eq. 10 implies the following,

$$0 \leq w'^{\top} x'^i + b \Rightarrow w'^{\top} x'^i + b + w_d < 0$$
$$0 > w'^{\top} x'^i + b \Rightarrow w'^{\top} x'^i + b + w_d \geq 0 \tag{11}$$

Combining the two equations in eq. 11 we get,

$$0 \leq w'^{\top} x'^i + b < -w_d$$
$$0 < -\left(w'^{\top} x'^i + b\right) \leq w_d \tag{12}$$

Note that the model (defined by $w, b$) is fixed. Hence the variables in the above conditions are the inputs $x$. Note that only one of the conditions in eq. 12 can be satisfied at any time, based on whether $w_d \geq 0$ or $w_d < 0$. The fraction of the inputs which satisfy one of the above conditions correspond to the fraction of responsive pairs and hence is the value of the score function.

Conditions in eq. 12 correspond to intersecting halfspaces formed by parallel hyperplanes. If $\|\bar{x}\|_2 \leq r$, the region of intersection can be upper bounded by a hypercuboid of length $2r$ in $d - 2$ dimensions and perpendicular length between the two hyperplanes $l = \frac{|w_d|}{\|w'\|_2}$ in the $(d-1)$-th dimension.

Hence, we can upper bound score function $s(h)$ as,

$$s(h) \leq \frac{(2r)^{d-2} \cdot l}{V_{d-1}(r)} \tag{13}$$

where $l = \frac{|w_d|}{\|w'\|_2}$ and $V_{d-1}(r)$ is the volume of the $(d-1)$-dimensional ball given by $\frac{\pi^{(d-1)/2}}{\Gamma\left(\frac{d-1}{2}+1\right)} r^{d-1}$ and $\Gamma$ is Euler's Gamma function.

Upon simplification we get,

$$s(h) \leq \frac{2^{d-2}}{\pi^{\left(\frac{d-1}{2}\right)}} \frac{l}{r \Gamma\left(\frac{d+1}{2}\right)} \tag{14}$$

Assuming $r = 1$ and $\|w'\|_2 = 1$, we can write eq. 14 as,

$$s(h) \leq \text{C} \cdot |w_d| \tag{15}$$

where $\text{C} = \frac{2^{d-2}}{\pi^{\left(\frac{d-1}{2}\right)} \cdot \Gamma\left(\frac{d+1}{2}\right)}$ is a constant for small dimensions.

$\square$

## A.4 Auditing Linear Classifiers with Anchor Explanations

**Imperfect Precision.** When the precision is not perfect, the anchor augmentation scheme (lines 11-13 in Alg. 2) is not as straightforward. Essentially we cannot assign the same label to all randomly sampled points in the hyperrectangle. To overcome this problem we can use some heuristics and tricks. (a) Note that we wish to exploit the information given by anchors to automatically label samples, but due to imperfect precision there will be errors in this labeling if we use our old augmentation scheme – this is analogous to active learning with noisy labels and we can explore existing literature in this field to deal with this problem. (b) We can internally consider a smaller hyperrectangle than that supplied by the data scientist and only sample from that – this can reduce the error arising from potential wrong labeling (c) We can also ask for labels for some points within each anchor - this might help due to the structure of linear classifiers (only points closer to the decision boundary can have imperfect precision) and the fact that the problem only arises when the precision is somewhere in between 0 and 1, it does not arise at the ends or closer to 0 or 1. Imperfect precision can potentially increase the number of queries or the computations required for anchor augmentation.

Alabdulmohsin et al. (2015) proposed a query synthesis spectral algorithm to learn homogeneous linear classifiers in $O(d \log \frac{1}{\Delta})$ steps where $\Delta$ corresponds to a bound on the error between estimated and true

classifier. They maintain a version space of consistent hypotheses approximated using the largest ellipsoid $\varepsilon^\star = (\mu^\star, \Sigma^\star)$ where $\mu^\star$ is the center and $\Sigma^\star$ is the covariance matrix of the ellipsoid. They prove that the optimal query which halves the version space is orthogonal to $\mu^\star$ and maximizes the projection in the direction of the eigenvectors of $\Sigma^\star$.

We propose an auditor $\texttt{AlgLC}_a$ as depicted in alg. 2 using their algorithm. The anchor explanations are incorporated through anchor augmentation. But, in the worst-case anchors are not helpful and hence the algorithm reduces essentially to that of Alabdulmohsin et al. (2015) (without anchors). In this section we find the query complexity of this auditor.

**Notation** In $\texttt{AlgLC}_a$, $\varepsilon^{t\star} = (\mu^{t\star}, \Sigma^{t\star})$ denotes the largest ellipsoid that approximates the version space (corresponds to search space in our case) at time $t$ where $\mu^{t\star}$ is the center and $\Sigma^{t\star}$ is the covariance matrix of the ellipsoid at time $t$. $N^t$ is the orthonormal basis of the orthogonal complement of $\mu^{t\star}$ and $N^{t'}$ is its transpose. $\alpha^{t\star}$ is the top eigenvector of the matrix $N^{t'}\Sigma^{t\star}N^t$. In the implementation by Alabdulmohsin et al. (2015), some warm-up labeled points are supplied by the user, we denote this set as $W$. Let $W(t)$ denote the $t$-th element of this set. Let the $d^{th}$ feature be the feature of interest without loss of generality.

---

**Algorithm 9** $\texttt{AlgLC}_a$ : Auditing Linear Classifiers using Anchors

---

1: **Input:** $T$, augmentation size $s$, set of warm-up labeled points $W$
2: set of queried points $Q := \emptyset$, $l :=\text{size}(W)$
3: **for** $t = 1, 2, 3, \ldots, T + l$ **do**
4:    **if** $t <= l$ **then**
5:       $(x, y) := W(t)$
6:       $Q \leftarrow Q \cup (x, y)$
7:       goto step 15
8:    **else**
9:       Query point $x^t := N^t\alpha^{t\star}$ from the DS
10:      Auditor receives label $y$ and explanation $A_x$ from the DS
11:      $Q \leftarrow Q \cup (x^t, y)$
12:      Sample randomly $q$ points $x^{t1} \ldots x^{tq}$ from $A_x$     } Anchor Augmentation
13:      $Q \leftarrow Q \cup \{(x^{t1}, y) \ldots (x^{tq}, y)\}$
14:    **end if**
15:    $\varepsilon^{(t+1)*} = \left(\mu^{(t+1)\star}, \Sigma^{(t+1)\star}\right) := \text{estimate\_ellipsoid}(Q)$
16:    $N^{t+1} := \text{update\_N}(\mu^{(t+1)\star})$
17:    $\alpha^{(t+1)\star} := \text{update\_alpha}(N^{t+1}, \Sigma^{(t+1)\star})$
18:    $\{//\text{Exact formulae for steps 15, 16 and 17 can be found in Alabdulmohsin et al. (2015)}\}$
19: **end for**
20: $\hat{w} = \mu^{T+1}$
21: **if** $|\hat{w}_i| \leq \Delta$ ($i^{th}$ feature is the FoI) **then**
22:    return No
23: **else**
24:    return Yes
25: **end if**

---

Next we give a bound on the number of queries required to audit using $\texttt{AlgLC}_a$. With worst-case anchors, it means that we are just using the algorithm of Alabdulmohsin et al. (2015), essentially without explanations and anchor augmentation. The auditor has a fixed $\epsilon$ that it decides beforehand. $\texttt{AlgLC}_a$ decides how many times it must run the algorithm of Alabdulmohsin et al. (2015) such that for the fixed $\epsilon$, it satisfies def. 2.

**Theorem 3.3.** *For every dimension $d$, there exists $c > 0$ such that for any $\epsilon \in (0, 1)$, auditor $\texttt{AlgLC}_a$ is an $(\epsilon, 0)$-auditor for feature sensitivity and $\mathcal{H}_{LC}$ with $T = O\left(d \log \frac{2c}{\epsilon}\right)$ queries.*

*Proof.* Let $w$ be the true classifier and $\hat{w}$ be the estimated classifier learnt by $\texttt{AlgLC}_a$.

Let the difference between $w$ and $\hat{w}$ be bounded by $\Delta$ as follows,

$$\|w - \hat{w}\|_2 \leq \Delta. \tag{16}$$

The value of $\Delta$ will be set later on. Since $\texttt{AlgLC}_a$ uses $|\hat{w}_d|$ to make its decision, the worst case is when the entire error in estimation is on the $d^{th}$ dimension. Hence, we consider $|w_d - \hat{w}_d| \leq \Delta$.

To guarantee that the auditor is an $(\epsilon, 0)$-auditor we need to verify for every hypothesis in the class that if $s(\cdot) = 0$, then the answer is No and if $s(\cdot) > \epsilon$, then the answer is Yes, see def. 2.

Importantly, $s(w)$ is zero only if $w_d = 0$ from theorem 1. If $w_d = 0 \Rightarrow |\hat{w}_d| \leq \Delta$, by eq. 16. Since $\texttt{AlgLC}_a$ returns a No for $|\hat{w}_d| \leq \Delta$, $\texttt{AlgLC}_a$ satisfies def. 2 when $s(w) = 0$ with $\delta = 0$.

Next, we have the case $s(w) > \epsilon$ when auditor should return a Yes with high probability. $\texttt{AlgLC}_a$ returns a Yes when $|\hat{w}_d| > \Delta$. Hence for $\texttt{AlgLC}_a$ to be correct, we need that $s(w) > \epsilon$ imply that $|\hat{w}_d| > \Delta$.

We can upper bound $s(w)$ using eq. 14 as,

$$s(w) \leq \textsc{c} \cdot |w_d| \tag{17}$$

Since $\epsilon < s(w)$,

$$\epsilon < \textsc{c} \cdot |w_d| \tag{18}$$

Since $|w_d - \hat{w}_d| \leq \Delta$,

$$\epsilon \leq \textsc{c} \left( |\hat{w}_d| + \Delta \right) \tag{19}$$

On rearranging,

$$\frac{\epsilon}{\textsc{c}} - \Delta \leq |\hat{w}_d| \tag{20}$$

Eq. 20 connects $\epsilon$ with $\Delta$ and $\hat{w}_d$. For $s(w) > \epsilon$, $|\hat{w}_d|$ should be greater than $\Delta$ for $\texttt{AlgLC}_a$ to be correct. Hence, lower bounding the LHS of eq. 20 we get,

$$
\begin{aligned}
\Delta &\leq \frac{\epsilon}{\textsc{c}} - \Delta \\
\Delta &\leq \frac{\epsilon}{\textsc{c}} - \Delta \\
\Delta &\leq \frac{\epsilon}{2\textsc{c}}
\end{aligned} \tag{21}
$$

We set $\Delta = \frac{\epsilon}{2\textsc{c}}$.

From Lemma 1, $\texttt{AlgLC}_a$ reduces to Alabdulmohsin et al. (2015)'s spectral algorithm in the worst-case. This algorithm has a bound of $O(d \log(\frac{1}{\Delta}))$. Hence, the query complexity of $\texttt{AlgLC}_a$ is $O(d \log(\frac{2\textsc{c}}{\epsilon}))$.

$\square$

## A.5 Auditing Decision Trees

---

**Algorithm 10** findpath$(x, P)$

---

1: Given : Query $x$ and explanation paths set $P$
2: **for** path $p \in P$ **do**
3:    $not\_p := False$
4:    **for** $(f_i, v_i, dir_i) \in p$ **do**
5:       **if** $(dir_i == \leq \& x_{f_i} > v_i) || (dir_i == \geq \& x_{f_i} < v_i)$ **then**
6:          $not\_p := True$
7:          goto line 3
8:       **end if**
9:    **end for**
10:   **if** $not\_p == False$ **then**
11:      return `True` {Query satisfies a pre-existing path}
12:   **end if**
13: **end for**
14: return `False`

---

**Algorithm 11** perturb$(x, p)$

---

1: Given : Query $x$ and corresponding explanation path $p = [(f, v, dir = \{\leq / \geq\})]$ {Two directions of inequality are used for ease of illustration. The code is generalizable to include other conditions easily.}
2: Randomly pick a tuple $(f, v, d)$ from $p$
3: **if** $dir == \leq$ **then**
4:   Perturb $x_f$ such that $x_f > v$
5: **else**
6:   Perturb $x_f$ such that $x_f < v$
7: **end if**
8: $x := x_{\backslash x_f} \cup x_f$
9: return $x$

---

**Theorem A.1.** *For any $\epsilon \in [0, 1]$, auditor* `AlgDT` *is an ($\epsilon$,0)-auditor for feature sensitivity and hypothesis class $\mathcal{H}_{DT}$ with $T = O(V)$ queries where $V$ is the number of nodes in the decision tree.*

*Proof.* Let $L$ be the number of leaves and $V$ be the number of nodes in the decision tree. The total number of queries asked by 3 equals the number of paths in the tree. The number of paths in a binary decision tree equals the number of leaves $L$ and $L = (V + 1)/2$. Once the auditor makes $V$ queries, it has explored all the paths in the tree and hence knows the tree exactly. Therefore with $O(V)$ queries, it can give the correct auditing decision precisely. $\square$

## A.6 Manipulation-Proofness

Yan and Zhang (2022) define manipulation-proofness as follows. Given a set of classifiers $V$, a classifier $h$, and a unlabeled dataset $S$, define the version space induced by $S$ to be $V(h, S) := \{h' \in V : h'(S) = h(S)\}$. An auditing algorithm is $\epsilon$-manipulation-proof if, for any $h^*$, it outputs a set of queries $S$ and estimate $\hat{s}$ that guarantees that $\max_{h \in \mathcal{V}(h^*, S)} |s(h) - \hat{s}| \leq \epsilon$.

Our anchor and decision tree auditors are manipulation-proof trivially since the version space size at the end of auditing is 1. Our counterfactual auditor is manipulation-proof since only those classifiers which have $w_i = 0$ will be in the version space.

### A.7 Experiments

For Adult dataset, the output variable is whether Income exceeds $50K/yr. For Covertype, the output variable is whether forest covertype is category 1 or not. For Credit Default, the output is default payment (0/1).

