# OpenReview forum: "XAudit : A Learning-Theoretic Look at Auditing with Explanations"
_TMLR — Accepted by TMLR_

### Review · Reviewer_btzA · 2024-03-18

**Summary Of Contributions:**

Strength:
- the learning theory approach to auditing fairness is new and useful
- the authors provide algorithms and characterize worst-case query complexity in a variety relevant settings, in particular i) linear classification ii) decision lists and iii) active learning
- The paper provides useful experiments, in particular those that also show average-case query complexity and complement the worst-case query complexity results

**Audience:**

Yes

**Claims And Evidence:**

Yes

**Requested Changes:**

- I would like to see more discussion towards my biggest concern, which is the existence of responsive pairs for all features. If I am thinking of the features as resulting from some causal graph, changing one feature causes changes in other features, and there may not be such a clean way to compare two individuals.
- I think the explanation of why the second algorithm for linear classifiers is useful can be improved (I am still not sure what the value of the anchor explanation based algorithm is when it has higher query complexity than the original algorithm the authors propose)

**Strengths And Weaknesses:**

Strengths:

Weaknesses:
- My biggest concern is that a lot of the paper seems to rely on the existence of "responsive pairs". For example, according to the authors' motivation that different features are correlated, it may be possible sometimes that these pairs don't exist for some features (for example, maybe I cannot change my gender and keep all the other features constant). What happens in this case/can the authors provide more motivation here?
- It is a bit disappointing that the auditing for active learning only works for finite hypothesis classes, and that for infinite classes the fallback is to basically solve the learning task. It would be nice to understand whether there is a gap between auditing and learning in general.

---

> ### Author Response · Authors · 2024-04-22
> **Author Response**
>
> Dear Reviewer,
>
> Thank you for taking the time to review our paper and providing us with your valuable feedback.
>
> We are glad that you find our learning theoretic approach to auditing new and useful, like our experiments and appreciate the variety of relevant settings used in our paper.
>
> Next we address your concerns.
>
> **Responsive pairs** : We use membership query active learning in our paper, where the queries are artificially constructed. Whether these queries exist in real life or not is not necessarily a concern. However, your point of using a causal graph is a very good one and can reduce the input space, leading to faster auditing. This is a great avenue for future research. We are happy to add this discussion to our paper.
>
> **Active Learning for infinite classes** : For infinite hypothesis classes, we can use Algorithm 4 (algorithm for finite hypothesis classes) but with different conditions for picking the next query and stopping conditions. The main issue in the picking_next_query function (Algorithm 6 in the appendix) is that there are infinite elements in the search space now and hence cardinality cannot be used as a measure of the search space – in this case we suggest to approximate the search space with a high dimensional geometric object (such as ellipsoid used in the anchors active learning algorithm since linear classifiers are also infinite dimensional) and use its volume as the size of the search space, |S|. For the stopping condition, we propose exploiting the structure of the hypothesis class and explanation methods to reduce the search space to a single hypothesis or to a set where the value of the score function is same for each of its elements, in which case similar stopping conditions as the finite case can be used. We have mentioned these in the first and third paragraphs of page 8.
>
> **Anchors** : The goal of the paper is to *analyze if and when explanations can lead to faster auditing*, for which we considered different kinds of explanations. Anchors give us a counter example where explanations do not lead to faster auditing – demonstrating this is the purpose of the anchors algorithm.
>
> Please let us know if you have any follow up questions or concerns.
>
> We look forward to hearing from you!
>
> Cheers,
>
> Authors

---

> ### Author Response · Authors · 2024-04-25
> **Check-in**
>
> Dear Reviewer,
>
> Hope you are doing well. We just wanted to check if you got a chance to read our response. We are looking forward to your reply and would be happy to address any remaining concerns.
>
> Cheers,
>
> Authors

---

> > ### Comment · Reviewer_btzA · 2024-04-28
> > **Sorry for the delay!**
> >
> > Apologies for the delay, it has been a bit of an insane week.
> >
> > - Regarding the anchor explanations, this does make sense to me.
> > - Can this be formalized? Also, I am expecting there would be computational complexity issues due to dealing with a high-dimensional grid; is there anything that can be done there?
> > - Responsive pairs: I am not quite sure I fully understand the response; could you please clarify? I understand that the queries are artificially constructed and this is fine. My worry is that whether there exists responsive pairs is not a function of the queries, but of the data itself. The data could be such that there is strong correlations between features, and you cannot change one feature without changing the others. In this case, you cannot isolate a single feature and change it while picking the rest of them constant.

---

> ### Author Response · Authors · 2024-04-29
>
> Dear Reviewer,
>
> Thanks for responding to our response, we really appreciate it!
>
> (1) Formalization for infinite hypothesis class : We believe any formalization or complexity bounds (beyond what we say in our response) are tightly coupled with the type of hypothesis class. Therefore, it is hard to do this with a sufficient generalization while still being useful. A lot of active learning literature is also class-specific. As an example, see [a] which we use in our paper, it proposes an algorithm for active learning linear classifiers with ellipsoidal approximations.
>
> (2) Responsive Pairs : Thanks for clarifying your question. Responsive pairs depend on the model. The way we have defined responsive pairs is as follows : First there is a set of all pairs and then the pairs which have differing predictions are called responsive pairs. Your question is about how the pairs are constructed. The pairs need not be constructed from a data distribution -- you are assuming a data distribution from which pairs are to be sampled, but the pairs/queries need not be from a distribution, they can be artificially synthesized like in membership query active learning and therefore the correlations according to the data distribution need not be respected.
>
> Incase pairs are created using a distribution (your comment) then we can essentially ignore the correlated feature while checking similarity. or consider the original and correlated feature together rather than just the original feature. This is feasible as we consider what features are correlated to be known to the auditor -- we believe this to be a reasonable assumption as auditing is not done is isolation and just like model developers have to do data pre-processing before building the model, correlation analysis should be a homework that auditors should do. There could also be regulators which supply this information to the auditors or could be public knowledge from research studies. Note that the auditor knows what the features are apriori.
>
> Hope this made sense.
> Please let us know if you have remainder concerns!
>
> Cheers,
>
> Authors
>
> *****************
>
> [a] Efficient active learning of halfspaces via query synthesis. Alabdulmohsin I., et. al. 2015

---

> > ### Comment · Reviewer_btzA · 2024-04-29
> >
> > I see, I do understand a bit better now, this seems to make sense. Thank you for you quick response!

---

### Review · Reviewer_Gc1M · 2024-03-19

**Summary Of Contributions:**

This paper explores a theoretical framework for the auditing of ML models when explanations are provided. They consider 3 types of explanations, across linear models and decision trees, and focus specifically on feature sensitivity, giving worst-case query complexity analysis. They empirically show that the average case is often better than the worst-case, and discuss some approaches for verifying that model owner outputs are truthful in the auditing process.

**Audience:**

Yes

**Broader Impact Concerns:**

no concerns

**Claims And Evidence:**

No

**Requested Changes:**

See the Strengths and Weaknesses section for details. In summary:

Key Weakness to address:
- paper scoping: content is much narrower than what is discussed, mostly due to focus on a strong form of feature sensitivity

Further: needs many clarifications across theoretical and experimental sections - it's unclear to me how many of these are critical weaknesses. But a quick list of some that jump out:
- more details on what decision tree path explanations look like
- make explicit if there are negative tests in the experiments
- things around formulation of the auditing setup
- more details on how the anchor setup works

**Strengths And Weaknesses:**

Strengths:
-The approach taken in this paper is fresh and original, attempting to understand the theoretical utility of various forms of explanations
- Auditing is an important area that does not receive enough attention - I think definitions like \eps-\delta auditors are useful pieces for this to be studied further

Critiques:
- I think the scope of this paper is narrower than the framing makes it seem - there is a specific focus on feature sensitivity and I’m not convinced that much of the reasoning applies to other metrics. For instance, distributional metrics (e.g. fairness metrics, subgroup accuracy) have very different properties for which I don’t see similar analogues to what is presented in the paper, which relies on very specific properties of feature sensitivity.
- Additionally, a very strong form of feature sensitivity (*any* sensitivity, globally) is considered, which confuses me a bit - the score function is given as the pct of pairs that are responsive, but this is never actually used. It’s not clear how much of the work applies to weaker, more practical notions of sensitivity
- Due to the highly specific nature of a lot of these things, it’s unclear to me how helpful the theoretical results are, given the very general framing of the paper
- On feature sensitivity: the example given is pen markings on an image, but I don’t think the method can be applied here - this is an abstract property that is emergent from many input features. A more relevant example might be helpful
- the focus is only on binary classifiers, whereas the framing is much more general - not clear how most of this applies to continuous or structured output
- It’s not clear to me what the path explanations in decision trees actually look like - what is contained in each node? Should be explained in more detail. Relatedly, what is “consistency” with a given decision path?
- I’m not sure there’s that much insightful information in the section on Stopping Conditions: it seems like in the finite case, the approach involves enumerating all remaining options, and there is no general approach given for infinite spaces
- I’m unclear in the Experiments section: are the results for all audits “Yes”? As in, do all methods have feature sensitivity? If yes, then probably should do some tests for the negative case; if no, then should discuss how models that are insensitive were derived.

Questions - could use clarification in paper:
- Top of p3: why is auditor returning a binary response, rather than an estimate of s?
- Def 2: why is s < \epsilon not included here? I see the authors state that it would make guarantees harder, but it does seem like a useful property that at least merits some discussion
- P4, top: where does the big-O characterization in “A Simple Baseline” come from?
- In the proof of Thm 3.2, authors state that the auditor is deterministic - it’s not clear to me that Alg 1, line 1 is deterministic
- Should be more explanation of how anchors work - I get confused around which A_x is returned - is this usually specified?
- Should discuss how important the simplification of perfect precision for anchors is
- Not clear to me why in 5.1 we check for faster learning, rather than faster auditing
- How is the worst case line in Fig 1 calculated?



Small points:
- Could be some more precise language in the intro (“societal applications”, “what an audit really signifies”)
- I read the proof of Thm 3.2 in the Appendix and it makes sense - it would be nice to give a little intuition in the main body
- Middle of p5: what is the “search space”?
- What is the dimensionality of the 3 datasets used?
- In 5.1, how are the hyperrectangles chosen? It seems non trivial given the precision constraints - can’t always choose the hyperrectangle of volume V centered at a point x

---

> ### Author Response · Authors · 2024-04-22
> **Response**
>
> Dear Reviewer,
>
> Thank you for taking the time to review our paper and providing us with your valuable feedback.
>
> We are very glad that you find our approach fresh and original, consider the problem of auditing important and think that the definitions introduced by our paper are useful.
>
> We agree to add clarifications for your minor points in the paper. Next we address your major concerns.
>
> (1) **Aim & scope of the paper** : The aim of our paper is to develop a theoretical framework for auditing & understand if and when explanations can help audits. We use feature sensitivity as an example to demonstrate our definitions, develop auditing algorithms and gain some understanding regarding the theoretical frameworks for auditing and utility of explanations. As such, the auditing algorithms and query complexities derived for feature sensitivity have to be specific to feature sensitivity, and this will perhaps be the case for any auditing property – in order to get good auditors, we have to look at them on a case-by-case basis. Having said that, there do exist a lot of general things from our paper that carry over to other auditing properties and settings. Namely,
>
> (a) We provide a negative result that explanations may not always reduce the query complexity of auditing – this is quite counterintuitive, as explanations supposedly give additional information and have a lot of structure to them. We also show some instances where explanations do help a lot. This means that whether explanations will help or not should be analyzed on a case-by-case basis.
>
> (b) We provide a very unique perspective to auditing by making a connection to active learning – not trying to blow our own horn, but to be honest, there was no prior work to the best of our knowledge (other than a concurrent work) which looked at auditing with this lens. Such a perspective unveils a very interesting insight – that any active learning algorithm can be used to audit – first learn the hidden model through active learning and then estimate the value of the score function on that learnt model. On the other hand, we also show that auditing is a simpler task than active learning and therefore we should look for smarter algorithms as well.
>
> (c) We formalize the process of auditing and give important definitions which can serve as a foundation for different variants. This was previously only considered an empirical field.
>
> (d) We give general auditing algorithms which can be applicable elsewhere.
>
> (2) **Generalization of our framework to distribution-dependent properties** : Thank you for pointing out distribution dependent properties. Our framework also applies to these. For distribution-dependent properties the score function should also depend on the distribution. The auditor will need to know the said distribution or there will be error if it uses a different distribution – this error term can be analyzed based on the difference between the two distributions.
>
> The general idea to audit a distribution-dependent property like demographic parity under our framework is
>
> 1. Find the hidden classifier through active learning queries.
> 2. Next, estimate the score function for the learnt classifier by sampling from the said distribution.
>
> Or for finite hypothesis class, it can also use the following:
>
> 1. Ask an active learning query to the DS
> 2. DS gives label to the auditor
> 3. Auditor updates its search space of hypotheses
> 4. Auditor estimates s(.) for all hypothesis in the search space (using samples from the said distribution) and checks if s(.)>$\epsilon$ for all hypothesis in the search space → return Yes and if s(.)<= $\epsilon$ for all hypothesis in the search space → return no. Else go to step 1.
>
> We will be very happy to add this discussion to our paper.
>
> (3) **Negative tests in the experiments** : The experiments are for the “Yes” case. We will make it explicit in the experiments section. We can add some results for “No” case but we dont believe that it makes much of a difference as : the query complexity for counterfactuals is fixed irrespective of yes or no; as  for anchors and decision trees, the auditors work by learning the hidden model first, in which case also it doesn’t matter whether there is feature sensitivity or not. Also the kind of questions we ask in the experiments section are relative in nature : typical vs worst-case anchors and active vs random testing queries for decision trees – therefore we don’t believe the “yes” or “no” case makes much of a difference here.
>
> (4) **Decision paths** : The nodes in an explanation path correspond to decision rules. For example, x_3 > 0.5 → x_2 < 0.9 → x_0 = “F” . Since the auditor already knows the input query, it can immediately verify that the input follows the path given as explanation.
> All the paths which lead to the same decision as returned by the auditor and respect the input features are consistent with the input and its decision. We will be happy to add this discussion to the paper.
>
> (contd.)

---

> ### Author Response · Authors · 2024-04-22
>
> (continuation)
>
> (5) **Auditing setup** : Certainly, the auditor can return an estimate of the score function rather than a binary response. Though there is an advantages to framing the problem in a binary response way : we note that the auditor can stop when all the hypotheses left in the search space have s(.) > $\epsilon$ or <=  $\epsilon$ (see stopping condition page 8) – this means that the auditor can stop without knowing what the exact hidden hypothesis is and therefore need not know what is the exact value of the score function. This leads to faster auditing.
> Irrespective, the auditor can certainly return an estimate of the score function in other formulations.
>
> (6) **How anchors work** : We are not sure about your question for A_x but here is an answer to the best of our understanding of your question : An anchor A_x is just a hyperrectangle such that (i) it contains x and (ii) $\tau$ fraction of the points in A_x have the same label as x. This anchor is given to the auditor by the data scientist. How this anchor is found by the data scientist is an orthogonal question to the auditor and there could exist multiple ways of building it. It doesn’t matter whether this hyperrectangle is *centered* around x or not as long as x is contained inside it.
>
> (7) **Relaxation of perfect precision** : We believe that assuming a perfect precision especially for linear classifiers (which is what we consider) may not be a deal breaker for the following reasons.
>
> Firstly, for linear classifiers, most points do have a perfect precision, it is just the points near the decision boundary which can have imperfect precision (this can be seen by drawing a linear classifier in 2D).
>
> When the precision is not perfect, the anchor augmentation scheme (lines 11-13 in Alg. 2) is not as straightforward. Essentially we cannot assign the same label to all randomly sampled points in the hyperrectangle. To overcome this problem we can use some heuristics and tricks like using literature from active learning with noisy labels , auditor internally considering a smaller hyperrectangle than that supplied by the data scientist and only sample from that, using distance between the asked query and the sampled points from the anchor and the direction of this distance vector to guide which samples to consider etc.
>
> (8) **Qs regarding Experiments on anchors** : Learning instead of auditing - For anchors, the auditor basically learns the hidden model first using active learning. That is why we use learning and auditing interchangeably here.
>
> Worst case line in fig 1 : For worst-case, anchor explanations provide no more information than just using labels. Therefore we just use labels (and no explanations) for plotting the worst-case line. We use Alg. 2 but without anchors (as using anchors vs not using anchors is the same in the worst-case).
>
> (9) **Simple Baseline Query Complexity** : Here is the simple theorem and its proof.
>
> Thm : For $\epsilon, \delta \in (0,1)$, auditor is an ($\epsilon,\delta$)-auditor if it queries $m \geq \frac{1}{\epsilon}\log \frac{1}{\delta}$ pairs.
>
> Proof:  Let the set of all pairs be denoted by $P$. Assume the fraction of responsive pairs is at least $\epsilon$. Then the probability that a randomly drawn pair is not a responsive pair is given by, $\Pr_{p_{ij}\sim P}(p_{ij}\; \text{ is not a responsive pair}) \leq 1-\epsilon$.
> If $m$ pairs are drawn randomly, then the probability that all of the pairs are not responsive is bounded by $(1-\epsilon)^m \leq \exp(-\epsilon m)\leq\delta$, by the choice of $m$.
>
> **********
>
> Please let us know if you have any follow up questions or concerns. We would be happy to address them.
>
> Looking forward to hearing from you!
>
> Cheers,
>
> Authors

---

> ### Author Response · Authors · 2024-04-25
> **Check-in**
>
> Dear Reviewer,
>
> Hope you are doing well. We just wanted to check if you got a chance to read our response. We are looking forward to your reply and would be happy to address any remaining concerns.
>
> Cheers,
>
> Authors

---

> > ### Author Response · Authors · 2024-05-03
> >
> > Dear Reviewer,
> >
> > Hope you are doing well.
> >
> > Really sorry to bother you again. We totally understand that with some decisions out, conferences going on and deadlines coming up, its a crazy time right now. We just wanted to check if you are satisfied with our response or would like some further clarifications. We would to very happy to resolve them.
> >
> > Cheers,
> >
> > Authors

---

> > > ### Comment · Reviewer_Gc1M · 2024-05-06
> > > **Response**
> > >
> > > Hi there - apologies for the delay on my end. I'm still getting used to the cadence of TMLR and have not been as on top of it as I would like to be.
> > >
> > > I appreciate the rebuttal. My concerns about scoping remain, as I find the results actually given in the paper to be significantly narrower than the introduction suggests. I acknowledge that there is interesting work in here but not sure that I would say the paper provides "evidence to support its claims".
> > >
> > > Some more specific notes:
> > > - on distribution dependent metrics: the approach you suggest in the rebuttal is consistent with what the paper suggests. however, I find it a little odd that to audit say, demographic parity, one first needs to learn the underlying classifier. Couldn't we just query directly through the DS? what is gained by learning the underlying classifier first? Is it more efficient, robust? this type of approach is one reason why I'm not sure there's much actionable generalizable insight from the cases studied in the paper to other cases in this framework.
> > > - on negative tests: to me, without negative tests in the experiments, then I don't believe that we can draw any strong conclusions about the empirical performance of these auditing methods. there are other things that we can learn of course (as you focus on currently) but it seems like demonstrating the model's ability to identify the negative case is important. I'm happy to be wrong about this but at the very least some discussion of why this isn't done might make this section stronger.

---

> > > > ### Author Response · Authors · 2024-05-11
> > > >
> > > > Dear Reviewer,
> > > >
> > > > Hope you are doing well!
> > > > We acknowledge and thank you for the efforts spent in reviewing our paper.
> > > >
> > > > Thanks for responding, below we address your concerns.
> > > >
> > > > 1. Distribution-dependent metrics. You are correct in hinting about additional advantages for not using random sampling. Essentially randomly sampling datapoints and using them to audit for say demographic parity, is not manipulation-proof while our algorithms given in the response are manipulation-proof. Simply put, manipulation-proofness means that the audit results hold even after the model is swapped as long as the swapped model gives consistent predictions (with the original model) for a particular set of queries. Since for our algorithms, either we learn the hidden model or all the leftover models in the search space have score functions below or above a threshold, our algorithms are manipulation-proof.
> > > >
> > > > For detailed discussion on why random sampling is not manipulation-proof, please refer to concurrent work [1] page 2 ‘Baseline:iid sampling’ section. We mention manipulation-proofness of our algorithms on page 8 of our paper, just above connections to active learning paragraph. The mathematical definition is in appendix A.6.
> > > >
> > > >
> > > > 2. Negative Tests : We have done some quick tests on the negative case for anchors. The results can be found here - https://anonymous.4open.science/r/rebuttal_response-CD2B/Results_Rebuttal.pdf (please download the pdf if you can't see the plots). We took the same classifiers as in the paper, but just made the weights zero for the feature of interest (foI). Then we run our experiments with and without anchors on these. We plot the error between the weight for the original foI weight (which is zero) and the estimated foI weight, since the anchor algorithm uses weight of the feature of interest to make a decision. As can be seen in the plots, we converge to a value very close to zero in all the cases, and with-anchor we converge to a more stable point quicker than without anchors.
> > > >
> > > > We are happy to include/highlight the current discussion in the paper.
> > > >
> > > > Hope you found our response valuable.
> > > >
> > > > We would be happy to address any remainder concerns. Looking forward to hearing from you!
> > > >
> > > > Cheers,
> > > >
> > > > Authors
> > > > **********
> > > >
> > > > [1] Active Fairness Auditing Yan et.al.

---

### Review · Reviewer_qtxJ · 2024-04-15

**Summary Of Contributions:**

The paper proposes a framework where an auditor, in addition to having black-box access to the model also has access to the explanations provided by the model. The paper proposes an approach to use explanations to lower the query complexity of auditing.

**Audience:**

Yes

**Broader Impact Concerns:**

Sufficiently addressed.

**Claims And Evidence:**

Yes

**Requested Changes:**

I think there are several ways that the clarifications can help the reader.

(1) What is an example of a non-testable auditing property?

(2) What is d in line 4 of algorithm 1?

(3) The score function $s$ does not necessarily reflect a statistical quantity, correct?

(4) Much of section 3.1.2 revolves around the assumption of perfect precision. This is very strong. Can this be relaxed?

(5) Why does feature sensitivity matter (in the way that it is defined)? Two features can be fully correlated.  Are there other interesting properties that I am overlooking?

(6) Why is the error between w and its estimation not converge to 0 in Fig 1?

**Strengths And Weaknesses:**

-- The idea of using an explanation for auditing and how it interacts with query complexity is novel.

-- A main weakness of the paper is the assumption that the data scientist is truthful.

-- Many details of the paper are unclear (see below).

---

> ### Author Response · Authors · 2024-04-22
>
> Dear Reviewer,
>
> Thank you for taking the time to review our paper and providing us with your valuable feedback.
>
> We are glad that you find the idea of using explanations for auditing and analyzing the query complexity to be novel.
>
> Next we address your concerns.
>
> **Non-testable Auditing Property** : Testable auditing property is defined using the outputs of hypotheses on an instance space. However, the score function depends on the hypothesis. So it could be that the outputs of two hypotheses are the same on the instance space used for auditing yet their decision boundaries are different. This leads to non-testable auditing properties.
> Consider two classifiers which agree on every query in the instance space. Yet their robustness to distribution shifts or adversarial attacks can be very different. We have the earlier discussion in the paper already and we are happy to add this example as well.
>
> **d in line 4 of algorithm 1** : Thanks for catching the typo. It is supposed to denote the feature of interest that we are auditing for. But this notation is overloaded and we will fix it.
>
> **Score function** : Yes that’s correct.
>
> **Relaxation of Perfect Precision** : We believe that assuming a perfect precision especially for linear classifiers (which is what we consider) may not be a deal breaker for the following reasons.
>
> Firstly, for linear classifiers, most points do have a perfect precision, it is just the points near the decision boundary which can have imperfect precision (this can be seen by drawing a linear classifier in 2D).
>
> When the precision is not perfect, the anchor augmentation scheme (lines 11-13 in Alg. 2) is not as straightforward. Essentially we cannot assign the same label to all randomly sampled points in the hyperrectangle. To overcome this problem we can use some heuristics and tricks. (a) Note that we wish to exploit the information given by anchors to automatically label samples, but due to imperfect precision there will be errors in this labeling if we use our old augmentation scheme – this is analogous to active learning with noisy labels and we can explore existing literature in this field to deal with this problem. (b) We can internally consider a smaller hyperrectangle than that supplied by the data scientist and only sample from that – this can reduce the error arising from potential wrong labeling (c) Use distance between the asked query and the sampled points from the anchor and the direction of this distance vector to guide which samples to consider. We can also ask for labels for some points within each anchor. This might help due to the structure of linear classifiers and the fact that the problem only arises when the precision is somewhere in between 0 and 1, it does not arise at the ends or closer to 0 or 1.
>
> Imperfect precision can potentially increase the number of queries or the computations required for anchor augmentation.
>
> **Correlations in Feature Sensitivity** : We wanted to build some theory for auditing and analyze if explanations can help us audit. To understand these, we use feature sensitivity to begin with and then go on to give general algorithms. Therefore feature sensitivity has dual purposes : (i) to help demonstrate, build theory and understanding of auditing + explanations and (ii) functional purpose in real life.
> Now coming to your concern (which is related to functionality), yes you are right, two features can be fully correlated. We assume that there is a third party, for eg. a regulator or a government entity, which guides what features are correlated and what features should be audited for. Since auditing doesn’t exist in isolation and has an ecosystem surrounding it (especially true for government entities and organizations conducting the audit), we believe that this may not be an unrealistic assumption. We are happy to add this discussion to the paper.
>
> **Error between w and its estimation** : Our anchors auditor uses an active learning algorithm given by [a] which approximates the search space of hypotheses using ellipsoids. We think the error may not be approaching zero due to two plausible reasons. Firstly, the ellipsoidal approximation may not be good enough and secondly, a lot more queries could be needed to approach zero. We will be happy to add this discussion to the paper.
>
> **Data Scientist being truthful** : We agree that in many cases the data scientist may not be truthful. Accordingly, we have dedicated a section in the paper to discuss this issue (section 6) wherein we suggest some strategies. Note that prior to our paper there was no theory of auditing or how explanations can help in auditing to the best of our knowledge. We have an exciting negative result even when the data scientist is truthful – that not all explanations can help lower the query complexity of auditing – we believe that this is very interesting and counterintuitive. Therefore the case of a truthful data scientist is also worth studying in our opinion. (cntd.)

---

> ### Author Response · Authors · 2024-04-22
>
> (continuation) That being said, we certainly agree that the case of an untruthful data scientist warrants much more research and is an exciting avenue for future research.
>
> Please let us know if you have any follow up questions or concerns.
>
> We look forward to hearing from you!
>
> Cheers,
>
> Authors
>
> **************
> [a]  Efficient active learning of halfspaces via query synthesis. Alabdulmohsin I., et. al. 2015

---

> ### Author Response · Authors · 2024-04-25
> **Check-in**
>
> Dear Reviewer,
>
> Hope you are doing well. We just wanted to check if you got a chance to read our response. We are looking forward to your reply and would be happy to address any remaining concerns.
>
> Cheers,
>
> Authors

---

> > ### Comment · Reviewer_qtxJ · 2024-04-28
> > **Response to the rebuttal**
> >
> > Thanks for the response and apologies for being slow at getting back to you.
> >
> > I think you can incorporate some of these questions and your responses in a future work section. I understand not all these requests can be done in the current paper.
> >
> > I am generally happy with all the responses. For estimating w, is there a way to get more insights on which of the reasons is more plausible?

---

> > > ### Author Response · Authors · 2024-04-29
> > > **Response**
> > >
> > > Dear Reviewer,
> > >
> > > Thanks for responding to our response, we really appreciate it!
> > >
> > > We are glad that you are satisfied with our responses. Yes, we agree, we can add some of the questions and our responses to the future works section.
> > >
> > > Regarding estimating w -- we believe the more likely reason is the ellipsoid approximation as we ran the algorithm for a larger number of queries in the past but the error was still there.
> > >
> > > Please let us know if you have remainder concerns!
> > >
> > > Cheers,
> > >
> > > Authors

---

### Decision · Action_Editor_FDEB · 2024-06-13

**Recommendation:** Accept with minor revision

**Comment:**

The reviewers have provided constructive feedback to the authors, raising concerns about the existence of responsive pairs for all features, the limitations of the auditing for active learning in the case of infinite hypothesis classes, and gaps in the experimental setup. The authors have responded to these concerns in a satisfactory manner, acknowledging the limitations of their approach and suggesting avenues for future research. All of this should be included during the revisions, and the scope of the paper made clear in the abstract and introduction. Overall, the paper presents an interesting and novel theoretical framework to auditing fairness in machine learning models. The findings of the paper are likely to be of interest to researchers and practitioners in the field of machine learning.

**Audience:**

The paper addresses an important topic in machine learning, namely auditing fairness in machine learning models. The reviewers also acknowledge that the paper’s findings may be of interest to the community, stating that the learning theory approach to auditing fairness is useful and novel. While the scope is limited, the paper might provide a foundation for a fairly novel direction for others to build on.

**Claims And Evidence:**

The authors provide theoretical analysis, algorithms, and empirical results to support their claims. The reviewers acknowledged that the key strengths of the paper are the new learning theory approach to auditing fairness, and the characterization of worst-case query complexity. The framing, however, suggests stronger results and broader scope. The authors should consider adjusting their abstract and introduction to accurately present the actual contributions, acknowledging the gaps. Further, the experiments, as noted by the reviewers, are somewhat weak. Some of the additional experiments done during the rebuttal should be included in the final submission as they strengthen the work.